# Disparities in ART Live Birth and Cumulative Live Birth Outcomes for Hispanic and Asian Women Compared to White Non-Hispanic Women

**DOI:** 10.3390/jcm10122615

**Published:** 2021-06-14

**Authors:** Alexander M. Kotlyar, Burcin Simsek, David B. Seifer

**Affiliations:** 1Department of Obstetrics, Gynecology and Reproductive Sciences, Yale School of Medicine, New Haven, CT 06510, USA; david.seifer@yale.edu; 2Department of Statistics, University of Pittsburgh, Pittsburgh, PA 15260, USA; burcinsim@gmail.com

**Keywords:** racial or ethnic disparities, Hispanic, Asian, non-Hispanic white, live birth rate, cumulative live birth rate, mandated states, non-mandated states, outcomes research

## Abstract

BACKGROUND: Conflicting disparities have been seen in assisted reproductive technology (ART) outcomes for Hispanic and Asian women compared to white, non-Hispanic (WNH) women. We, therefore, sought to clarify these disparities and calculated cumulative live birth rates (CLBR) for these racial or ethnic groups using the SARTCORS database. METHODS: We performed an analysis of the 2014–2016 SARTCORS database for member clinics doing at least 50 cycles of ART each year. RESULTS: In comparison to cycles in WNH women, cycles in Hispanic and Asian patients were in older (*p* < 0.001), more nulliparous women, that were less likely to have a history of endometriosis compared WNH women regardless of prior ART status. ART cycles in Hispanic and Asian women, exhibited lower rates of live birth (LB) per cycle start (*p* < 0.001) compared to cycles in WNH women. Multivariate logistic regression demonstrated that cycles from Hispanic and Asian women were less likely to have a LB and CLBR than white women (OR 0.86; *p* = 0.004, OR 0.69; *p* < 0.001, respectively) independent of age, parity, BMI, etiology of infertility, use of ICSI or number of embryos transferred. CONCLUSIONS: Race or ethnicity continues to be an independent prognostic factor for LB and CLBR for ART. Additional analysis of trends among Hispanic and Asian women is warranted to enable addressing disparities in outcomes in ART treatment.

## 1. Introduction

Racial and ethnic disparities in infertility and infertility treatment outcomes are well established [1]. Numerous studies have observed disparities in outcomes between black non-Hispanic (BNH) women and white non-Hispanic (WNH) women in regards to their success using assisted reproductive technologies (ART) for autologous, fresh, non-donor cycles. These disparities include lower clinical intra-uterine gestation (CIG), live birth rate (LBR) per transfer, and LBR per cycle start. Three large database studies have compared outcomes between cycles from BNH and WNH women utilizing the Society for Assisted Reproductive Technologies Clinical Outcome Reporting System (SARTCORS) and have demonstrated race as a robust independent predictor of live birth outcomes in ART cycles [1,2,3].

Although clear disparities have been consistently seen between BNH women and WNH women, the degree of disparity is much less clear for Hispanic and Asian women. A single-center study from 2011 comparing 134 Hispanic patients to 301 WNH patients showed no difference in CIG and SAB rates nor was there any difference noted in LBR. However, they did note that Hispanic women were more likely to have tubal factor infertility and endometriosis [2]. These findings are in contrast to two large scale studies utilizing the SARTCORS database showing lower LBR and higher spontaneous abortion (SAB) rates [3,4]. Additional studies from outside the US also showed a lower LBR in Asian women than white-Caucasian women undergoing fresh embryo transfer [5]. Several other single-center studies did not find a difference in CIG rate and LBR, even after controlling for confounding variables such as age, BMI, day 3 FSH, smoking status, and infertility diagnosis [6,7,8]. Therefore, ongoing reporting discrepancies exists concerning ART outcomes in Hispanic and Asian women. 

A similar inconsistent situation is noted with women identifying themselves as Asian. Several earlier SARTCORS database studies showed lower clinical pregnancy rates and LBR in Asian compared to WNH women [3,4,9]. However, the most contemporary of these studies examined a cohort over 10 years ago from 2004–2006. In addition, several single-center studies report conflicting results with some showing a lower LBR in Asian women compared to WNH and others showing no difference [7,10,11].

Nearly 10 years have passed since the question of outcome disparities has been addressed between Hispanic or Asian women and their WNH counterparts. Furthermore, no study has yet to determine if there are disparities in cumulative live birth rate (CLBR) as prior studies have focused only upon the initial cycle that women underwent followed by their immediate fresh transfer [12]. In addition, the question of whether state-mandated insurance coverage has influenced disparities among Hispanic and Asian patients has yet to be examined. We, therefore, sought to address these questions by performing an analysis of ART cycle outcomes in Asian and Hispanic women using the 2014–2016 SARTCORS database.

## 2. Materials and Methods

### 2.1. Data Source and Inclusion Criteria

This was a retrospective cohort study utilizing the 2014–2016 SARTCORS dataset. This study was deemed exempt from review by the institutional review board of the Yale School of Medicine as the dataset was anonymous and de-identified. De-identified data from member clinics comprising over 91% of reported ART cycles from 2014–2016 from the SARTCORS database were extracted. This data had been validated by SART and were also reported to the Centers for Disease Control as part of the Fertility Clinic Success Rate and Certification Act of 1992. Data fields were validated with 10 of 11 sampled data fields showing discrepancy rates of 5% or less. The data submitted from member clinics included information about ART treatment cycles and the outcomes of both fresh and frozen autologous cycles, donor cycles, and non-donor embryo transfer cycles based upon a standardized protocol. Race was designated as white non-Hispanic (WNH), Hispanic, Asian, Black non-Hispanic (BNH), Native American or other which is inconsistent with Centers for Disease Control (CDC) definitions of racial categories [13].

The initial SARTCORS dataset was obtained from the SARTCORS data vendor, RedShift Technologies. This initial dataset contained 563,730 ART cycles during the 2014–2016 time period. In total, 1753 cycles were excluded since they were from clinics performing less than 50 cycles per year to avoid sampling bias. A total of 219,171 (39%) of cycles were excluded from the resultant 561,977 cycles due to missing data on race or ethnicity. Of note, a comparison was done between the 219,171 excluded cycles and the original 561,977 cycles, and the LBR was same in both confirming that this exclusion did not skew the data. In total, 162,632 of the 561,977 original cycles were fresh autologous cycles. A total of 11,530 fresh cycles (6.5%) were subsequently excluded since donor oocytes were used. Another 2530 fresh autologous cycles (0.02%) were excluded since race or ethnicity were reported in multiple categories to minimize confounding. In total, 11,710 non-WNH, non-Hispanic, non-Asian cycles were also excluded. Therefore, 148,572 fresh autologous cycles were included in the study (See Figure 1). Our analysis also assessed cumulative live birth rates (CLBR) which were obtained by linking all embryo transfers to their original index oocyte retrieval cycles for cycles in 2014 and 2015. Our CLBR analysis included 12,885 cycles which had been reported for Hispanic women, 26,683 cycles which had been reported for Asian women, and 109,004 cycles for NHW women. A subgroup analysis cycles was performed in states with an insurance mandate which was defined as states mandating third party-payer coverage of ART for the study period of 2014–2016. These mandated states included Arkansas, Connecticut, Hawaii, Illinois, Maryland, Massachusetts, Rhode Island, and New Jersey.

### 2.2. Statistical Analysis

We analyzed the data using R 3.5.1 package for Windows (Microsoft, Redmond, WA, USA). Our unit of analysis was the treatment cycle given that the data were de-identified. Data from cycles in which the women had no prior ART cycle were analyzed separately since these patients may have a more favorable prognosis given lack of prior failed ART cycle. Diagnoses were assessed separately, i.e., if a patient had more than one etiology of infertility, each diagnosis was analyzed individually. Unusually high (outliers) FSH dosage values (e.g., >80 ampules) were removed since they may have been coding errors. Implantation rate was determined by obtaining the quotient of the number of fetal heartbeats in a given cycle and the number of embryos transferred in the cycle. We defined clinical pregnancy rates as the presence of a gestational sac seen via first-trimester ultrasound. Live birth was defined as the birth of one or more living infants. Rates of each of these parameters were determined per cycle start. 

The definition of a cumulative live birth rate (CLBR) was set as the birth of at least one living infant from an associated primary transfer (fresh or frozen and thaw (FET)). SART defines CLBR as the number of embryo transfers associated or linked with a source index retrieval cycle within one year of that retrieval cycle. One limitation of the database is that it does not include start dates for cycles or dates of retrievals and transfers. However, “reporting year” (the year for cycle start) for each cycle is listed in the database. Therefore, we calculated the cumulative rate within a 24-month maximum timeframe. Additional details on how the CLBR was calculated can be found in Seifer et al. [14].

Only two-tailed statistical tests were used with a *p*-value of <0.05 being considered significant. Percentages did not equal to 100 due to rounding, and there were different numbers of cycles in some analyses because of missing data. Chi-squared testing was used for categorical variables. Student’s *t*-test was used for continuous variables; however, if the distributions were non-normal, a Mann-Whitney test was used. The 95% confidence intervals were reported for all values. To determine the contribution of race or ethnicity on ART treatment outcomes, multivariable logistic regression analyses were performed by adjusting for potential confounding factors including age, parity, BMI, etiology of infertility, use of ICSI, and number of embryos transferred.

## 3. Results

In this study, we analyzed the disparities between Hispanic and Asian women compared to WNH women. Our analysis included 8341 cycles from Hispanic women with no prior ART, and 4544 cycles with Hispanic patients that did undergo prior ART. In total, 14,696 cycles in Asian women with no prior ART and 11,987 cycles in women that did undergo prior ART were analyzed. The Hispanic and Asian cycles with no prior ART were compared to 64,878 cycles in WNH women with no prior ART. For the Asian and Hispanic cycles with prior ART, they were compared to 44,126 cycles from WNH women with prior ART (Table 1 and Table 2).

We note clear disparities in age and BMI. For both Hispanic and Asian women, cycles from both women involved significantly older women compared to cycles involving WNH women (*p* < 0.001). This difference was sustained regardless of whether the cycles were in women with or without prior ART. Asian women tended to have a lower BMI and were more likely to be nulliparous whether they had or had not had prior ART compared to WNH women (*p* < 0.001). Examining cycles from Hispanic women with no prior ART showed greater nulliparity compared to WNH women (*p* < 0.001). (Table 1 and Table 2). The opposite BMI trend was seen in Hispanic women (*p* < 0.001).

Concerning etiology of infertility, cycles from Hispanic and Asian women showed substantial divergence. Although cycles from Asian women with tubal factor was not significantly different compared to WNH women regardless of ART history, tubal factor was more prevalent among cycles from Hispanic compared to WNH women in populations with and without prior ART (Table 1 and Table 2). Cycles from Asian and Hispanic women were also noted to have a higher prevalence of uterine factor (*p* < 0.001) and diminished ovarian reserve (DOR) (*p* < 0.001), but a lower risk of male factor (*p* < 0.001) and endometriosis (*p* < 0.001) compared to WNH women. Notably, no difference was seen in the proportion of patients with unexplained infertility among Asian women and WNH women. However, unexplained infertility was more prevalent among cycles from WNH women compared to Hispanic women (*p* < 0.001). Overall, while cycles from Hispanic women were noted to have the greatest prevalence of tubal factor among the three racial or ethnic groups, both Hispanic and Asian cycles were characterized by greater uterine factor and DOR as their underlying infertility etiologies compared to WNH women.

Substantial disparities were also noted in ART cycle outcomes. Cycles from Asian women with no prior ART were noted to have greater rates of cycle cancelation (*p* = 0.01) compared to WNH women; however, this was not seen in women with prior ART. ICSI was utilized for more cycles from Asian women compared to WNH women regardless of prior ART usage (*p* < 0.001). Interestingly, Asian women tended to have a significantly greater number of cycles in which 1 embryo or 3 or more embryos were produced (*p* < 0.001). Despite this, implantation rates were significantly lower for cycles from Asian women among both the prior and no prior ART groups. In addition, Asian cycles exhibited broadly lower ART success rates with lower clinical intrauterine gestation (CIG) rates, higher spontaneous abortion (SAB) rates, and lower LBR per CIG, as well as LBR per cycle start compared to WNH cycles. No difference was seen in ectopic and heterotopic rates (Table 1 and Table 2) between the three groups of women. Aside from higher cycle cancelation rates in women with prior ART, cycles from Hispanic women exhibited similar disparities in outcomes as Asian women compared to WNH women. Of note, Hispanic women had cycles which led to greater production of 2 or more embryos compared to WNH women (*p* < 0.001). Yet, ART outcomes were not improved despite a slightly higher number of embryos transferred for women with and without prior ART (Table 1 and Table 2).

To assess if race or ethnicity had an independent effect on ART outcomes, we performed a multivariable regression analysis for both Hispanic and Asian cycles with and without prior ART. For Hispanic women, race remained an independent predictor of live birth resulting from an ART cycle with a lower chance of live birth for cycles from Hispanic women compared to cycles from WNH women (OR 0.86, 95% CI 0.77–0.95 for women without prior ART (initial cycle), OR 0.80, 95% CI 0.70–0.91 for women with prior ART (Table 3)). This finding was independent of age, BMI, cause of infertility, history of past spontaneous abortions, use of ICSI and number of embryos transferred. When examining cycles from Asian women, we noted similar results with Asian ethnicity as being an independent predictor of live birth regardless whether or not the cycle was associated with prior ART (OR 0.68, 95% CI 0.62–0.75 for women without prior ART (initial index cycle), OR 0.81, 95% CI 0.74–0.88 for women with prior ART (Table 3)).

We next examined cumulative live birth rates (CLBR) for cycles in Hispanic and Asian women compared to WNH women for those with and without prior ART. For Hispanic women, race remained a predictor of a lower cumulative live birth rate compared to WNH even when controlling for age, parity, history of spontaneous abortions, cause of infertility, diminished ovarian reserve, Day 3 FSH, AMH, ICSI, and number of embryos transferred (OR 0.84, 95% CI: 0.77–0.92) (Table 4). A similar situation was observed for cycles from Asian women when controlling for the same confounders showing Hispanic ethnicity was associated with a lower cumulative live birth rate compared to cycles from WNH women (OR 0.79, 95% CI: 0.71–0.85, *p* < 0.001) (Table 4).

## 4. Discussion

In this study, we assessed the ART outcomes of cycles from Hispanic and Asian women in the 2014–2016 SARTCORS dataset. Compared to cycles from WNH women, cycles from Hispanic women exhibited a greater proportion of tubal factor and a greater frequency of uterine factor and DOR compared to cycles from WNH women. Cycles from both ethnic or racial groups showed lower proportions of CIG rate and LBR compared to cycles from WNH women. In addition, cycles from Hispanic and Asian women exhibited significantly lower CLBR in contrast to WNH women. These differences were independent of age, BMI, cause of infertility, past spontaneous abortions, use of ICSI and number of embryos transferred. This study represents the largest analysis of ART cycle outcomes from Hispanic and Asian women compared to cycles from WNH women to present. In addition, it is the first such study to analyze CLBR between all three groups.

The disparities noted in the data of ART cycle outcomes for Asian and Hispanic women are likely to be multifactorial. This may be secondary to differences in access-to-care and socio-economic status (SES). Both racial or ethnic groups we examined in the current study have a lower SES compared to WNH patients [15]. Lower SES is a well-established factor that is associated with poorer health outcomes [16]. In addition, cycles from both Hispanic and Asian women were overall older and more nulliparous than cycles from WNH women. Although this in-of-itself can lead to a lower LBR given the rapid decline of ovarian reserve with advancing age, AMH also tends to be lower for Asian and Hispanic women when stratified for age compared to WNH, especially at younger-to-middle ages [17,18,19]. Furthermore, the combination of older age and greater frequency of nulliparity suggests that Hispanic and Asian women are presenting for ART treatment at a later point in their life cycle for their initial pregnancy attempt as compared to WNH women. The social, cultural, and economic factors influencing these findings of older age and greater nulliparity require future study to determine a better understanding of the underlying causes. Of note, obesity has previously been shown to be associated with poorer ART outcomes in Hispanic patients and obese Asian patients [20]. Although the significantly greater proportion of obesity among Hispanic women in our population may be contributing to the lower LBR of those cycles without prior ART (initial cycles) and prior ART cycles, we observed a significantly lower percentage of obesity among Asian women. This suggests that despite the substantially healthier BMI profile of Asian women, this was not enough to offset the more deleterious influence of older age impacting the lower LBR compared to cycles from WNH women.

These data support several previously defined features of Hispanic and Asian women undergoing ART. Consistent with Fujimoto et al [6] and Shuler et al [4], tubal factor was noted to be greater among Hispanic women compared to WNH women [2,4]. This is consistent with prior work showing greater Chlamydia infection rates among Hispanic women with an adjusted odds ratio (aOR) of 1.46 (95% CI: 1.33–1.61) compared to WNH women [21]. One notable difference is that our study did find a lower LBR for cycles in Hispanic women compared to WNH women which Shuler et al did not [2]. This difference may be secondary to the smaller sample size resulting in less power to determine a real difference and an increased possibility of a Type 2 error when examining 134 Hispanic and 301 WNH women involved in the latter study.

When examining the clinical characteristics of ART cycles for both Hispanic women and Asian, stark contrasts are noted. In cycles in women without prior ART, the percentage of elective single embryo transfer (eSET) was significantly lower among Hispanic women (16.1%) compared to WNH women (23.9%). However, no difference was seen for cycles in Asian women compared to WNH women without prior ART. This difference confirms data from a prior SARTCORS database analysis in 2010 showing higher percentages of eSET among Asian women compared to WNH and lower eSET rates in Hispanic women [22]. Such consistency indicates that over the subsequent six years the difference in performance of eSET has persisted. One possible explanation for the difference in eSET is the proportion of women with AMH levels less than 1 ng/mL in women less than 40 years of age which is consistent with diminished ovarian reserve. Among all three racial or ethnic groups, this proportion was lowest among cycles in Asian women and the highest among Hispanic women representing an inverse relationship between these variables.

A key novel feature of this study was the analysis of linked cycles for 2014 and 2015 that involved primary transfers of either fresh or frozen and thawed embryos from a single linked retrieval. This is notable in the CLBR among all age groups older than 35 years from cycles from Asian women and Hispanic women compared to WNH women. Throughout all ages, CLBR was lower in cycles in Hispanic and Asian women compared to WNH women. Furthermore, this disparity widened with increasing age thus emphasizing the lower prognostic outlook for non-white women in older age categories.

The presence of state insurance mandates for ART coverage may be one possible explanation for contributing to these disparities. For patients without prior ART in mandated states, the percentage of cycles were 74.6% for WNH women, 8.2% for Hispanic women, and 17.1% for Asian women. In non-mandated states this distribution was similar with 73.3% of cycles in WNH women, 10.0% in Hispanic women, and 16.7% in Asian women. When looking at patients with prior ART cycles in mandated states, the percentage of cycles were 75.5% for WNH women, 7.2% for Hispanic women, and 17.3% for Asian women. Although for patients with prior ART in non-mandated states this distribution was comparable with 70.5% of cycles in WNH women, 7.6% in Hispanic women, and 21.9% in Asian women. We can place these percentages in context when we consider the racial distribution of mandated states with 69.5% of women being WNH, 14.1% being Hispanic, and 9.7% being Asian. For non-mandated states, the racial distribution is 81.2% WNH, 12.1% Hispanic, and 3.7% Asian [23]. Therefore, in mandated states, Asians tended to utilize ART less than expected for their portion of the population and WNH women and Hispanic women tend to utilize ART more than expected. As for non-mandated states, both Hispanic and Asians used ART more than expected for their portion of the population. Hence the presence of an insurance mandate did not seem to enhance ART use in the Hispanic and Asian populations. This is surprising given the greater use of ART for other racial groups such as black women in mandated states compared to non-mandated states in comparison to NHW [14]. Furthermore, prior work assessing ART access via non-SARTCORS databases has observed greater numbers of ART cycles in Hispanic and Asian women in mandated states [24]. Considering that prior work has shown no difference in infertility rates between races or ethnicities and that insurance access does affect utilization of ART, the presence of an insurance mandate for ART does not seem to be a modifiable factor to improve ART access for Hispanic and Asian women [25,26].

The reasons for the surprising lack of increased ART utilization in mandated states among Hispanic and Asian patients may be multi-fold. First studies of socioeconomic status distribution among various racial and ethnic groups indicate that Hispanic and Asian women have a socioeconomic status that is intermediate between WNH women and black women [15]. Hence, there could be a threshold SES beyond which a state insurance mandate may not enhance access to ART treatments. Second, even with the presence of an insurance mandate, the structure of coverage for infertility treatment in a state may bias a patient towards more conservative treatments such as ovulation induction with or without intra-uterine insemination depending on the cost-effectiveness assessment by the insurance company within that state [27].

Several strengths and limitations are inherent to the SARTCORS database. First, this database is the largest de-identified, standardized, and validated source of ART data collected from clinics throughout the US. Not only does this enhance the statistical power of this study, it also enhances the generalizability of these findings given the varied ethnic and racial make-up of the United States. Furthermore, this database compiles data from a nationwide network of ART clinics over the span of decades which also allows for identification of and analysis of trends in ART practice and outcomes. Despite the above advantages, a weakness is the lack of information on socioeconomic status of the patients undergoing these ART cycles. Markers of socioeconomic status, such as annual salaries or surrogate indicators such as a highest obtained educational level, remain a key confounder when assessing the impact of race or ethnicity on ART outcomes. An additional limitation of the database is the presence of missing data. In total, 39% of cycles reported in SARTCORS did not have data identifying race or ethnicity even though it is a required field. Race itself a self-reported field which may become more subjective as the population of the US becomes more heterogeneous over time. Furthermore, aside from the general race categories which are consistent with CDC race definitions, there is no further subdivision with each group, i.e., Asians being separated into Chinese, South Asian, South-East Asian, etc. Such subdivisions also exhibit differences in ART outcomes [28].

Despite these acknowledged limitations these data indicate that race or ethnicity remains an independent factor of outcome and access to ART in the US for Hispanic and Asian women. These data further suggest that several factors may have contributed to these disparities and that efforts can be made to mitigate these factors. Such mitigating efforts could involve enhancing education on the age-based decline in fertility to various racial or ethnic groups with consideration to seeking care at a younger age when initially trying to conceive, adequate healthy diet and exercise to maintain a normal BMI, in addition to an expansion of available insurance coverage. Utilizing these efforts may eventually narrow and, ideally, eliminate the ART outcome gaps noted in Hispanic and Asian women.

## Figures and Tables

**Figure 1 jcm-10-02615-f001:**
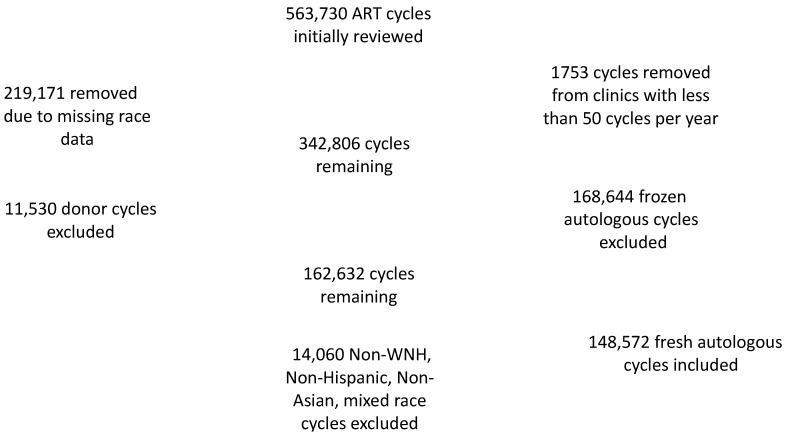
Scheme for SARTCORS database cycle inclusion.

**Table 1 jcm-10-02615-t001:** Baseline characteristics, treatment, and outcomes for fresh, nondonor cycles among Hispanic, Asian, and white women with no prior AT.

	No Prior ART						No Prior ART				
	Hispanic (*n* = 8341)		White (*n* = 64,878)				Asian (*n* = 14,696)		White (*n* = 64,878)		
Characteristics (% Reporting)	%	95% CI	%	95% CI	*p*	Characteristics (% Reporting)	%	95% CI	%	95% CI	*p*
Women’s age (year)					<0.001	Women’s age (year)					<0.001
<35	43.4	(42.3−44.5)	54.6	(54.2−55)		<35	42.7	(41.9−43.5)	54.6	(54.2−55)	
35−37	22.6	(21.7−23.5)	21.2	(20.8−21.5)		35−37	22.6	(21.9−23.3)	21.2	(20.8−21.5)	
38−40	20.1	(19.2−20.9)	15.2	(14.9−15.4)		38−40	19	(18.4−19.7)	15.2	(14.9−15.4)	
41–42	8.5	(7.9–9.1)	5.7	(5.5–5.8)		41–42	8.7	(8.2–9.1)	5.7	(5.5–5.8)	
>42	5.5	(5.0–6.0)	3.5	(3.3–3.6)		>42	7	(6.6–7.4)	3.5	(3.3–3.6)	
Nulliparous (49.6)	43.3	(41.8–44.8)	51.2	(50.5–51.8)	<0.001	Nulliparous (48.6)	55.4	(54.1–56.8)	51.2	(50.5–51.8)	<0.001
Past spontaneous abortions	24	(23.1–24.9)	20.2	(19.9–20.5)	<0.001	Past spontaneous abortions	19	(18.4–19.7)	20.2	(19.9–20.5)	0.001
Diagnosis						Diagnosis					
Tubal Factor	25.6	(24.7–26.6)	11	(10.8–11.2)	<0.001	Tubal Factor	10.5	(10.0–11.0)	11	(10.8–11.2)	0.076
Tubal Ligation	8	(7.4–8.6)	1.7	(1.6–1.8)	<0.001	Tubal Ligation	0.6	(0.5–0.7)	1.7	(1.6–1.8)	<0.001
Hydrosalpinx	1.5	(1.3–1.8)	0.8	(0.7–0.8)	<0.001	Hydrosalpinx	1.1	(0.9–1.3)	0.8	(0.7–0.8)	<0.001
Other	16.4	(15.6–17.2)	8.6	(8.4–8.9)	<0.001	Other	9	(8.5–9.4)	8.6	(8.4–8.9)	0.192
Male Infertility	32.6	(31.6–33.6)	34.2	(33.9–34.6)	0.002	Male Infertility	29.1	(28.4–29.8)	34.2	(33.9–34.6)	<0.001
Uterine Factor	5.8	(5.3–6.3)	3.8	(3.6–3.9)	<0.001	Uterine Factor	5.1	(4.8–5.5)	3.8	(3.6–3.9)	<0.001
History of endometriosis	7.4	(6.8–8.0)	9.2	(9–9.4)	<0.001	History of endometriosis	6.9	(6.5–7.3)	9.2	(9–9.4)	<0.001
Diminished ovarian reserve	26.1	(25.2–27.1)	21.5	(21.2–21.8)	<0.001	Diminished ovarian reserve	28.5	(27.7–29.2)	21.5	(21.2–21.8)	<0.001
Unexplained	8.3	(7.7–8.9)	13.3	(13–13.6)	<0.001	Unexplained	13.3	(12.8–13.9)	13.3	(13–13.6)	0.963
Ovulation Disorder (PCOS)	12.3	(11.6–13.0)	13.9	(13.7–14.2)	<0.001	Ovulation Disorder (PCOS)	11.6	(11.1–12.2)	13.9	(13.7–14.2)	<0.001
Other	19.6	(18.7–20.4)	25.5	(25.1–25.8)	<0.001	Other	31.4	(30.7–32.2)	25.5	(25.1–25.8)	<0.001
Day 3 FSH <10 IU/L	3.7	(3.4–4.2)	4	(3.8–4.1)	0.293	Day 3 FSH <10 IU/L	3.7	(3.5–4.1)	4	(3.8–4.1)	0.172
Day 3 FSH >10 IU/L	69	(68.0–70.0)	65.4	(65.1–65.8)	<0.001	Day 3 FSH >10 IU/L	67.8	(67.0–68.5)	65.4	(65.1–65.8)	<0.001
FSH dosage ≥ 37 ampules (93.7)	54.1	(52.9–55.2)	52.2	(51.8–52.6)	0.002	FSH dosage ≥ 37 ampules (93.5)	53.4	(52.6–54.3)	52.2	(51.8–52.6)	0.007
AMH <1 among women < 40 yr	13.6	(12.8–14.4)	13.4	(13.1–13.6)	0.616	AMH <1 among women < 40 yr	11.8	(11.2–12.4)	13.4	(13.1–13.6)	<0.001
High ovarian response	0.3	(0.2–0.5)	0.5	(0.4–0.5)	0.213	High ovarian response	0.4	(0.3–0.6)	0.5	(0.4–0.5)	0.761
Cycle Cancelled	10.9	(10.2–11.6)	9.9	(9.6–10.1)	0.004	Cycle Cancelled	10.6	(10.1–11.1)	9.9	(9.6–10.1)	0.01
Due to low response	82.4	(80.0–84.8)	84.1	(83.2–85.0)	0.188	Due to low response	84.5	(82.6–86.3)	84.1	(83.2–85.0)	0.735
ICSI	62.8	(61.8–63.9)	61.2	(60.8–61.6)	0.005	ICSI	57	(56.2–58.7)	61.2	(60.8–61.6)	<0.001
No of embryos					<0.001	No of embryos					<0.001
1	27.8	(26.6–29.0)	38.2	(37.7–38.6)		1	42	(40.9–43.1)	38.2	(37.7–38.6)	
2	59.4	(58.1–60.7)	54.1	(53.6–54.6)		2	47	(45.9–48.1)	54.1	(53.6–54.6)	
3+	12.8	(11.8–13.8)	7.7	(7.5–8)		3+	11	(10.3–11.7)	7.7	(7.5–8)	
No. of embryos transferred: mean (sd)	1.88 (0.71)		1.72 (0.67)		<0.001	No. of embryos transferred: mean (sd)	1.73 (0.79)		1.72 (0.67)		0.153
Implantation rate %: mean (sd)	69.71 (34.16)		76.52 (33.4)		<0.001	Implantation rate %: mean (sd)	74.26 (35.67)		76.52 (33.4)		0.001
Treatment outcome						Treatment outcome					
Clinical intrauterine gestation (CIG)	30.2	(29.2–31.2)	33	(32.7–33.4)	<0.001	Clinical intrauterine gestation (CIG)	22.5	(22.5–33.0)	33	(32.7–33.4)	<0.001
Spontaneous abortion	17.3	(15.9–18.9)	14.5	(14.1–15.0)	0.089	Spontaneous abortion	17.4	(16.2–18.8)	14.5	(14.1–15.0)	<0.001
Live birth per CIG	80.6	(79.0–82.1)	84.6	(84.1–85.0)	<0.001	Live birth per CIG	81.2	(79.9–82.5)	84.6	(84.1–85.0)	<0.001
Biochemical pregnancy	5.4	(4.9–5.9)	5.7	(5.5–5.9)	0.333	Biochemical pregnancy	4.6	(4.3–5.0)	5.7	(5.5–5.9)	<0.001
Ectopic or heterotopic	0.6	(0.5–0.8)	0.5	(0.5–0.6)	0.382	Ectopic or heterotopic	0.5	(0.4–0.7)	0.5	(0.5–0.6)	1
Not preganant	54.2	(53.1–55.3)	47.2	(46.8–47.6)	<0.001	Not preganant	51.7	(50.9–52.5)	47.2	(46.8–47.6)	<0.001
Live birth per cycle started	24.3	(23.4–25.3)	27.9	(27.6–28.3)	<0.001	Live birth per cycle started	18.2	(17.6–18.9)	27.9	(27.6–28.3)	<0.001
Plurality of birth (24.6)					0.005	Plurality of birth (23.1)					<0.001
Singleton	74.2	(72.2–76.0)	76.9	(76.2–77.5)		Singleton	82.1	(80.6–83.5)	76.9	(76.2–77.5)	
Twins	23.9	(22.1–25.9)	21.9	(21.3–22.5)		Twins	16.7	(15.3–18.1)	21.9	(21.3–22.5)	
Triples or more	1.9	(1.4–2.6)	1.3	(1.1–1.5)		Triples or more	1.2	(0.8–1.7)	1.3	(1.1–1.5)	
eSET (%) < 38 y/o (44.7)	16.1	(15.2–17.1)	23.9	(23.5–24.2)	<0.001	eSET (%) < 38 y/o (42.8)	23.2	(22.4–24.1)	23.9	(23.5–24.2)	0.178
BMI ≥ 30	38.3	(37.3–39.4)	31.5	(31.1–31.9)	<0.001	BMI ≥ 30	7.3	(6.9–7.7)	31.5	(31.1–31.9)	<0.001
States					<0.001	States					0.749
Mandated	23.9	(23.0–24.8)	27.7	(27.4–28.1)		Mandated	28	(27.3–28.7)	27.7	0	
Non-mandated	75.9	(74.9–76.8)	71.5	(71.1–71.8)		Non-mandated	71.8	(71.1–72.5)	71.5	0	

**Table 2 jcm-10-02615-t002:** Baseline characteristics, treatment, and outcomes for fresh, nondonor cycles among Hispanic, Asian, and WNH women with prior ART.

	Prior ART						Prior ART				
	Hispanic (*n* = 4544)		White (*n* = 44,126)				Asian (*n* = 11,987)		White (*n* = 44,126)		
Characteristics (% Reporting)	%	95% CI	%	95% CI	*p*	Characteristics (% Reporting)	%	95% CI	%	95% CI	*p*
Women’s age (year)					<0.001	Women’s age (year)					<0.001
<35	23.6	(22.4–24.9)	32.7	(32.2–33.1)		<35	21.7	(20.9–22.4)	32.7	(32.2–33.1)	
35–37	22	(20.8–23.2)	21.9	(21.5–22.3)		35–37	18.6	(17.9–19.3)	21.9	(21.5–22.3)	
38–40	26.4	(25.1–27.7)	22.1	(21.7–22.4)		38–40	24.6	(23.8–25.4)	22.1	(21.7–22.4)	
41–42	15.1	(14.1–16.2)	11.8	(11.5–12.1)		41–42	16.5	(15.8–17.2)	11.8	(11.5–12.1)	
>42	13	(12.0–14.0)	11.7	(11.4–12)		>42	18.7	(18.0–19.4)	11.7	(11.4–12)	
Nulliparous (49.6)	45.7	(43.8–47.5)	44.4	(43.8–44.9)	0.188	Nulliparous (48.6)	50.9	(49.7–52.1)	44.4	(43.8–44.9)	<0.001
Past spontaneous abortions	34.4	(33.1–35.8)	32.8	(32.4–33.3)	0.028	Past spontaneous abortions	32.3	(31.5–33.2)	32.8	(32.4–33.3)	0.311
Diagnosis						Diagnosis					
Tubal Factor	22.6	(21.4–23.8)	10.4	(10.2–10.7)	<0.001	Tubal Factor	10.7	(10.1–11.2)	10.4	(10.2–10.7)	0.496
Tubal Ligation	5.3	(4.7–6.0)	1	(0.9–1.1)	<0.001	Tubal Ligation	0.5	(0.3–0.6)	1	(0.9–1.1)	<0.001
Hydrosalpinx	2	(1.6–2.5)	0.7	(0.6–0.8)	<0.001	Hydrosalpinx	1.3	(1.1–1.6)	0.7	(0.6–0.8)	<0.001
Other	15.6	(14.5–16.7)	8.8	(8.6–9.2)	<0.001	Other	9.2	(8.7–9.7)	8.8	(8.6–9.2)	0.27
Male Infertility	32.5	(31.2–33.9)	36	(35.5–36.4)	<0.001	Male Infertility	28.4	(27.6–29.2)	36	(35.5–36.4)	<0.001
Uterine Factor	6.8	(6.1–7.6)	4.5	(4.3–4.7)	<0.001	Uterine Factor	5.5	(5.1–6.0)	4.5	(4.3–4.7)	<0.001
History of endometriosis	8.8	(8.0–9.7)	9.8	(9.5–10.1)	0.033	History of endometriosis	7.5	(7.0–8.0)	9.8	(9.5–10.1)	<0.001
Diminished ovarian reserve	44.2	(42.8–45.7)	41.3	(40.8–41.7)	<0.001	Diminished ovarian reserve	52.4	(51.5–53.3)	41.3	(40.8–41.7)	<0.001
Unexplained	7.2	(6.5–8.0)	10.4	(10.1–10.7)	<0.001	Unexplained	9.8	(9.3–10.3)	10.4	(10.1–10.7)	0.059
Ovulation Disorder (PCOS)	9.6	(8.8–10.5)	10.2	(10.0–10.5)	0.183	Ovulation Disorder (PCOS)	7.4	(7.0–7.9)	10.2	(10.0–10.5)	<0.001
Other	18.3	(17.2–19.5)	21.2	(20.9–21.6)	<0.001	Other	23.5	(22.7–24.3)	21.2	(20.9–21.6)	<0.001
Day 3 FSH <10 IU/L	3.2	(2.7–3.8)	4	(3.8–4.2)	0.014	Day 3 FSH <10 IU/L	4	(3.7–4.4)	4	(3.8–4.2)	0.855
Day 3 FSH >10 IU/L	77.6	(76.4–79.0)	74	(73.6–74.4)	<0.001	Day 3 FSH >10 IU/L	76.5	(75.8–77.3)	74	(73.6–74.4)	<0.001
FSH dosage ≥ 37 ampules (93.7)	64.5	(63.0–66.0)	64.8	(64.3–65.2)	0.716	FSH dosage ≥ 37 ampules (93.5)	60.6	(59.6–61.5)	64.8	(64.3–65.2)	<0.001
AMH <1 among women < 40 yr	21.6	(20.1–23.2)	21	(20.6–21.5)	0.471	AMH <1 among women < 40 yr	18.6	(17.7–19.6)	21	(20.6–21.5)	<0.001
High ovarian response	1.1	(0.9–1.5)	0.7	(0.7–0.8)	0.005	High ovarian response	1.2	(1.1–1.5)	0.7	(0.7–0.8)	<0.001
Cycle Cancelled	15.3	(14.3–16.4)	13.3	(13.0–13.6)	<0.001	Cycle Cancelled	13	(12.4–13.6)	13.3	(13.0–13.6)	0.302
Due to low response	74.4	(70.9–77.6)	80.1	(79.1–81.1)	0.001	Due to low response	76.1	(73.9–78.2)	80.1	(79.1–81.1)	0.001
ICSI	60.9	(59.5–62.3)	63.9	(63.4–64.3)	<0.001	ICSI	58.5	(57.6–59.4)	63.9	(63.4–64.3)	<0.001
No of embryos					<0.001	No of embryos					<0.001
1	21.8	(20.2–23.4)	26.3	(25.8–26.8)		1	28.2	(27.0–29.4)	26.3	(25.8–26.8)	
2	55.7	(53.8–57.6)	53.1	(52.5–53.7)		2	48.3	(47.0–49.7)	53.1	(52.5–53.7)	
3+	22.5	(21.0–24.2)	20.6	(20.1–21.1)		3+	23.5	(22.3–24.6)	20.6	(20.1–21.1)	
No. of embryos transferred: mean (sd)	2.08 (0.86)		2.02 (0.88)		<0.001	No. of embryos transferred: mean (sd)	2.08 (1.03)		2.02 (0.88)		<0.001
Implantation rate %: mean (sd)	60.83 (33.08)		67.08 (34.1)		<0.001	Implantation rate %: mean (sd)	63.71 (36.73)		67.08 (34.1)		<0.001
Treatment outcome						Treatment outcome					
Clinical intrauterine gestation (CIG)	22.5	(21.3–23.8)	25.1	(24.7–25.5)	<0.001	Clinical intrauterine gestation (CIG)	15.6	(15.0–16.3)	25.1	(24.7–25.5)	<0.001
Spontaneous abortion	23.2	(20.6–25.9)	18.6	(17.8–19.3)	<0.001	Spontaneous abortion	23.8	(21.9–25.8)	18.6	(17.8–19.3)	<0.001
Live birth per CIG	74.9	(72.1–77.5)	80.5	(79.7–81.2)	<0.001	Live birth per CIG	75	(73.0–77.0)	80.5	(79.7–81.2)	<0.001
Biochemical pregnancy	4.9	(4.3–5.6)	5.7	(5.5–6.0)	0.025	Biochemical pregnancy	4.3	(3.9–4.6)	5.7	(5.5–6.0)	<0.001
Ectopic or heterotopic	0.6	(0.4–0.9)	0.5	(0.4–0.6)	0.533	Ectopic or heterotopic	0.4	(0.3–0.5)	0.5	(0.4–0.6)	0.109
Not preganant	62.8	(61.4–64.2)	58.3	(57.8–58.7)	<0.001	Not preganant	63.2	(62.3–64.0)	58.3	(57.8–58.7)	<0.001
Live birth per cycle started	16.9	(15.8–18.0)	20.2	(19.8–20.6)	<0.001	Live birth per cycle started	11.7	(11.2–12.3)	20.2	(19.8–20.6)	<0.001
Plurality of birth (24.6)					0.217	Plurality of birth (23.1)					0.019
Singleton	76.5	(73.4–79.5)	75.8	(74.9–76.6)		Singleton	79.2	(77.0–81.3)	75.8	(74.9–76.6)	
Twins	21.1	(18.4–24.2)	22.6	(21.8–23.5)		Twins	19.4	(17.4–21.6)	22.6	(21.8–23.5)	
Triples or more	2.3	(1.4–3.7)	1.6	(1.3–1.9)		Triples or more	1.4	(0.8–2.2)	1.6	(1.3–1.9)	
eSET (%) < 38 y/o (44.7)	10.4	(9.2–11.9)	13.9	(13.4–14.3)	<0.001	eSET (%) < 38 y/o (42.8)	13.2	(12.2–14.2)	13.9	(13.4–14.3)	0.22
BMI ≥ 30	39.9	(38.5–41.3)	34.5	(34.1–35.0)	<0.001	BMI ≥ 30	6.4	(6.0–6.9)	34.5	(34.1–35.0)	<0.001
States					0.001	States					<0.001
Mandated	30.1	(28.8–31.5)	32.4	(32.0–32.9)		Mandated	27.3	(26.5–28.1)	32.4	(32.0–32.9)	
Non-mandated	69.8	(68.4–71.1)	67	(66.5–67.4)		Non-mandated	72.6	(71.8–73.4)	67	(66.5–67.4)	

**Table 3 jcm-10-02615-t003:** Independent predictors of achieving live birth among Hispanic, Asian, and WNH women.

	Hispanic vs. White					Asian vs. White			
	Without Prior ART		With Prior ART			Without Prior ART		With Prior ART	
	Odds Ratio (95% CI)	*p*	Odds Ratio (95% CI)	*p*		Odds Ratio (95% CI)	*p*	Odds Ratio (95% CI)	*p*
Race					Race				
White	Reference		Reference		White	Reference		Reference	
Hispanic	0.86 (0.77, 0.95)	0.004	0.80 (0.70, 0.91)	0.001	Asian	0.68 (0.62, 0.75)	<0.001	0.81 (0.74, 0.88)	<0.001
Women’s age (y)					Women’s age (y)				
<35	Reference		Reference		<35	Reference		Reference	
35–37	0.71 (0.68, 0.75)	<0.001	0.76 (0.71, 0.81)	<0.001	35–37	0.73 (0.69, 0.77)	<0.001	0.77 (0.72, 0.82)	<0.001
38–40	0.46 (0.43, 0.49)	<0.001	0.50 (0.46, 0.54)	<0.001	38–40	0.48 (0.44, 0.51)	<0.001	0.50 (0.46, 0.54)	<0.001
41–42	0.23 (0.20, 0.26)	<0.001	0.27 (0.24, 0.31)	<0.001	41–42	0.23 (0.20, 0.26)	<0.001	0.28 (0.25, 0.31)	<0.001
>42	0.09 (0.07, 0.17)	<0.001	0.10 (0.08, 0.12)	<0.001	>42	0.09 (0.07, 0.12)	<0.001	0.09 (0.07, 0.11)	<0.001
Nulliparous	0.91 (0.85, 0.97)	0.003	0.77 (0.72, 0.82)	<0.001	Nulliparous	0.90 (0.85, 0.97)	0.003	0.77 (0.72, 0.82)	<0.001
Past spontaneous abortions	0.93 (0.87, 0.99)	0.027	1.01 (0.95, 1.08)	0.742	Past spontaneous abortions	0.95 (0.89, 1.01)	0.095	1.02 (0.95, 1.08)	0.62
Tubal factor	0.94 (0.84, 1.06)	0.348	1.04 (0.92, 1.18)	0.542	Tubal factor	0.85 (0.75, 0.96)	0.01	0.99 (0.88, 1.12)	0.932
Male factor	1.05 (1.00, 1.10)	0.042	1.05 (0.99, 1.11)	0.09	Male factor	1.04 (1.00, 1.09)	0.067	1.05 (0.99, 1.11)	0.082
Uterine factor	0.84 (0.75, 0.93)	0.001	0.98 (.86, 1.13)	0.822	Uterine factor	0.83 (0.75, 0.93)	0.001	0.95 (0.83, 1.08)	0.461
Diminished ovarian reserve	0.83 (0.78, 0.89)	<0.001	0.82 (0.77, 0.88)	<0.001	Diminished ovarian reserve	0.83 (0.77, 0.88)	<0.001	0.82 (0.77, 0.88)	<0.001
Day 3 FSH					Day 3 FSH				
<10 IU/L	Reference		Reference		<10 IU/L	Reference		Reference	
≥10 IU/L	0.91 (0.83, 1.01)	0.081	0.92 (0.80, 1.05)	0.198	≥10 IU/L	0.93 (0.84, 1.02)	0.127	0.88 (0.78, 1.00)	0.056
AMH	1.39 (1.30, 1.50)	<0.001	1.24 (1.14, 1.35)	<0.001	AMH	1.33 (1.24, 1.43)	<0.001	1.24 (1.15, 1.34)	<0.001
ICSI	0.96 (0.91, 1.01)	0.104	0.95 (0.88, 1.03)	0.217	ICSI	0.96 (0.91, 1.01)	0.087	0.95 (0.89, 1.02)	0.2
Cycle Cancelled	32.26 (15.01, 83.96)	<0.001	40.83 (8.5, 733.23)	<0.001	Cycle Cancelled	42.07 (18.59, 120.75)	<0.001	17.23 (6.01, 72.56)	<0.001
No of embryos					No of embryos				
1	Reference		Reference		1	Reference		Reference	
2	1.29 (1.23, 1.35)	<0.001	1.45 (1.36, 1.54)	<0.001	2	1.26 (1.21, 1.32)	<0.001	1.40 (1.31, 1.49)	<0.001
3+	1.14 (1.03, 1.25)	0.009	1.22 (1.11, 1.33)	<0.001	3+	1.13 (1.03, 1.24)	0.01	1.15 (1.06, 1.26)	0.001
BMI ≥ 30	0.76 (0.73, 0.80)	<0.001	0.85 (0.79, 0.90)	<0.001	BMI ≥ 30	0.76 (0.72, 0.80)	<0.001	0.86 (0.81, 0.92)	<0.001

**Table 4 jcm-10-02615-t004:** Cumulative live birth rate for primary transfer (fresh or thawed FET) in Hispanic, Asian, and WNH women for 2014–2015.

	Hispanic vs. White					Asian vs. White			
	Without Prior ART		With Prior ART			Without Prior ART		With Prior ART	
	Odds Ratio (95% CI)	*p*	Odds Ratio (95% CI)	*p*		Odds Ratio (95% CI)	*p*	Odds Ratio (95% CI)	*p*
Race					Race				
White	Reference		Reference		White	Reference		Reference	
Hispanic	0.82 (0.73, 0.91)	<0.001	0.87 (0.77, 0.99)	0.031	Asian	0.79 (0.74, 0.85)	<0.001	0.84 (0.77, 0.92)	<0.001
Women’s age (y)					Women’s age (y)				
<35	Reference		Reference		<35	Reference		Reference	
35–37	0.71 (0.67, 0.75)	<0.001	0.78 (0.73, 0.84)	<0.001	35–37	0.72 (0.68, 0.76)	<0.001	0.79 (0.74, 0.84)	<0.001
38–40	0.49 (0.46, 0.53)	<0.001	0.57 (0.53, 0.61)	<0.001	38–40	0.50 (0.47, 0.54)	<0.001	0.57 (0.53, 0.62)	<0.001
41–42	0.25 (0.22, 0.29)	<0.001	0.36 (0.32, 0.40)	<0.001	41–42	0.26 (0.23, 0.29)	<0.001	0.35 (0.32, 0.39)	<0.001
>42	0.09 (0.07, 0.11)	<0.001	0.16 (0.14, 0.19)	<0.001	>42	0.09 (0.07, 0.16)	<0.001	0.16 (0.13, 0.18)	<0.001
Nulliparous	0.92 (0.86, 0.99)	0.03	0.88 (0.82, 0.94)	<0.001	Nulliparous	0.92 (0.85, 0.99)	0.024	0.89 (0.84, 0.95)	<0.001
Past spontaneous abortions	1.02 (0.95, 1.10)	0.513	1.01 (0.95, 1.08)	0.732	Past spontaneous abortions	1.00 (0.93, 1.08)	0.948	1.01 (0.95, 1.08)	0.75
Tubal factor	0.87 (0.77, 0.98)	0.026	0.88 (0.79, 0.99)	0.998	Tubal factor	0.86 (0.75, 0.98)	0.026	1.00 (0.89, 1.12)	0.998
Male factor	0.94 (0.89, 0.98)	0.01	0.98 (0.93, 1.03)	0.034	Male factor	0.96 (0.91, 1.01)	0.099	0.99 (0.94, 1.05)	0.847
Uterine factor	1.03 (0.92, 1.15)	0.612	1.01 (0.89, 1.13)	0.925	Uterine factor	1.01 (0.91, 1.13)	0.811	1.02 (0.91, 1.14)	0.746
Diminished ovarian reserve	0.74 (0.69, 0.80)	<0.001	0.82 (0.77, 0.88)	<0.001	Diminished ovarian reserve	0.73 (0.68, 0.79)	<0.001	0.82 (0.77, 0.88)	<0.001
Day 3 FSH					Day 3 FSH				
<10 IU/L	Reference		Reference		<10 IU/L	Reference		Reference	
≥10 IU/L	0.92 (0.82, 1.03)	0.146	0.89 (0.78, 1.02)	0.093	≥10 IU/L	0.91 (0.81, 1.02)	0.098	0.90 (0.79, 1.03)	0.128
AMH	2.07 (1.91, 2.24)	<0.001	1.77 (1.63, 1.93)	<0.001	AMH	2.02 (1.87, 2.18)	<0.001	1.84 (1.70, 2.00)	<0.001
ICSI	1.14 (1.08, 1.21)	<0.001	1.35 (1.27, 1.44)	<0.001	ICSI	1.16 (1.10, 1.23)	<0.001	1.34 (1.25, 1.42)	<0.001
Cycle Cancelled	1.50 (1.02, 2.16)	0.032	0.29 (0.16, 0.49)	<0.001	Cycle Cancelled	1.59 (1.08, 2.29)	0.015	0.37 (0.21, 0.63)	<0.001
No of embryos					No of embryos				
1	Reference		Reference		1	Reference		Reference	
2	0.73 (0.70, 0.77)	<0.001	0.86 (0.81, 0.91)	<0.001	2	0.71 (0.68, 0.75)	<0.001	0.84 (0.79, 0.89)	<0.001
3+	0.43 (0.39, 0.48)	<0.001	0.50 (0.46, 0.54)	<0.001	3+	0.41 (0.37, 0.46)	<0.001	0.47 (0.44, 0.51)	<0.001
BMI ≥ 30	0.71 (0.67, 0.75)	<0.001	0.83 (0.78, 0.88)	<0.001	BMI ≥ 30	0.71 (0.67, 0.75)	<0.001	0.85 (0.79, 0.90)	<0.001

## Data Availability

The data that support the findings of this study are available from SARTCORS but restrictions apply to the availability of these data, which were used under license for the current study, and so are not publicly available. Data are however available from the authors upon reasonable request and with permission of SARTCORS.

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
