# Peer review of "Disparities in ART Live Birth and Cumulative Live Birth Outcomes for Hispanic and Asian Women Compared to White Non-Hispanic Women"

_jcm, 2021, doi:10.3390/jcm10122615_

Round 1

Reviewer 1 Report

The paper is well well written. The study is of large sample size, but the novelty is limited. However, it is worth publishing the data as it is a large study. 

In the introduction, there are a lot of short forms other than some standard ones and it is worth expanding and possibly use common terms - For eg: clinical intrauterine gestation (CIG), but may be better to use clinical pregnancy.

Table 1 and 2 are very large. It would be best to focus on some key variables rather than presenting all the data that are available. May be could put the large table as supplementary file. Further, rather than presenting as two separate tables side by side (ie comaprison of Hispanic vs white and asian vs white) why not put it as three columns altogether ie hispanic vs asian vs white.

Further in the results, do you have mean or median of continuous variables (eg: age) rather than making it into variables of different groups like age <35, 35-37 .......etc. and presenting as percentages.

Results section need to be considerably shortened to make it easy for the reader to understand the key data (ie main outcome measures like cumulative live birth rates, LBR per cycle etc) better.

Asian population - please describe different subgroups (ie chines, south east asians etc). Success vary depend on different subgroups (Reference - Maalouf et al 2017 BJOG using HFEA UK national database). If you don't have the data of different subgroups mention that as limitations in the discussion section

Discussion is also too long - Focus the key findings, the possible reasons for the difference in live births, any interventions that may may change outcome, strengths and weakness, future direction

Author Response

The paper is well well written. The study is of large sample size, but the novelty is limited. However, it is worth publishing the data as it is a large study.  – Thank you for the feedback.

In the introduction, there are a lot of short forms other than some standard ones and it is worth expanding and possibly use common terms - For eg: clinical intrauterine gestation (CIG), but may be better to use clinical pregnancy. – We greatly appreciate this feedback; however, many terms used in short form such as clinical intrauterine gestation (CIG) were used because of their prevalence in prior literature. Hence, we used these to maintain consistency.

Table 1 and 2 are very large. It would be best to focus on some key variables rather than presenting all the data that are available. May be could put the large table as supplementary file. Further, rather than presenting as two separate tables side by side (ie comaprison of Hispanic vs white and asian vs white) why not put it as three columns altogether ie hispanic vs asian vs white. – We thank the reviewer for this valuable feedback. While we agree that Tables 1 and 2 are large, they are presenting essential demographic data and cycle outcome data that is typically included in the main text body in similar publications such as references 4, 7, 14, and 17. To be consistent with the prior literature, we chose to include all of the data into two large combined tables.

Further in the results, do you have mean or median of continuous variables (eg: age) rather than making it into variables of different groups like age <35, 35-37 .......etc. and presenting as percentages. – We do have the mean age, however we chose to show the distribution among the various age groups both since we see this as more informative and to stay consistent with prior papers that also listed the age stratifications such as reference 14 and is consistent with the age stratification presented in the SARCORS database.

Results section need to be considerably shortened to make it easy for the reader to understand the key data (ie main outcome measures like cumulative live birth rates, LBR per cycle etc) better. – We have shortened the results section in particular the sections outlining the demographic and infertility cause data. Further abbreviation could not be possible without excluding essential findings in ART outcomes and in our multivariable analysis.

Asian population - please describe different subgroups (ie chines, south east asians etc). Success vary depend on different subgroups (Reference - Maalouf et al 2017 BJOG using HFEA UK national database). If you don't have the data of different subgroups mention that as limitations in the discussion section – Thank you for this feedback, we have incorporated this reference and stated explicitly in the limitations that the SARTCORS database does not have any further sub-divisions for Asian ethnicity.

Discussion is also too long - Focus the key findings, the possible reasons for the difference in live births, any interventions that may may change outcome, strengths and weakness, future direction – We have also shortened the discussion, namely the paragraph which now comprises lines 284-291. However, further abbreviation while still covering key findings, differences in demographics, ART outcomes, placing the findings in the context of prior research, interventions, strengths and weakness, and future directions in the context of the size and scope of the SARTCORS database and the multifactorial causes behind

racial disparities is not possible without substantially reducing the quality of the discussion section.

Reviewer 2 Report

Paper jcm-1213734 by Kotlyar et al.

1.             Title. Disparities in ART live birth and cumulative live birth outcomes for Hispanic and Asian women compared to white non-Hispanic women.

2.             Summary

This paper intends to analyze, from a national register, the impact of racial origin on infertility treatment (ART) results

3.    Overall opinion

The paper concerns a relatively interesting question, with an almost perfect methodology, impressive results and is well written. However, there may be several explanations on the underlying mechanism that may be related to confounders which could not be addressed in the study. Moreover, several critical points need to be addressed to increase the demonstration. The references are very USA oriented (see below) and need to be enriched. Some methodological points can be discussed.

However, after taking into consideration the reviewer’s remarks, the paper will surely deserves publication

4.    Introduction

  • Introduction is interesting, extended, with the potential mechanisms. However, many references are not very recent (11 of 27 before 2011), or rely on the same database (10 from SART), or from US data (n=24). Only 1 comes from outside and 2 from literature reviews. It is clear that the racial item is much more studied in USA than elsewhere, but the paper could be enriched with some other references as Wiltshire (2020), Chiang (2012), Mascarenhas (2019), Begum (2016), etc. could be included for discussion of racial differences.
  • Define SAB before first use (I suppose spontaneous abortion)

5.    Methods

  • Methods are extensively described and are an example of what has to be done in this matter. The inclusion/non-inclusion criteria are clear and a flow chart is given.
  • Race affiliation needs to be explained for non-American readers and, if possible, discussed
  • Apparently, all initially included cycles were fresh and FET cycles (563730 cycles), and , then, only fresh cycles were included, merged with FET cycles issued from the finally included fresh cycles to make possible the cumulative rates calculations. This is ok but needs to be better explained. Moreover, it is necessary to explain why the merge was not done for 2016 fresh cycles, and how long after the index aspiration could the FET cycles could be merged (1, 2 years or more or only during the period 2014-2016).
  • Was the implantation rate only computed for the fresh cycle ?
  • Confounders: The most important were included, but a couple of ones that may have some relation with both racial category and outcome variable need to be at least discussed or included: economic status, distance from the ART centre. Finally, there is a potential bias if WNH people are of higher economic level and, thus, may access to centres with higher technicity. It may be difficult to analyse this factor, but that may play a role in the racial differences shown by authors. It is at least necessary to analyse a centre effect

6.    Results

Results are well expressed. Tables 1 and 2 give a lot of statistically significant differences. However, there were apparently many covariables introduced in the multivariate analyses. The text is ambiguous between Material / methods and results sections

7.    Discussion

Discussion is well written, relatively extended. However, it is not easy to find a real explanation of the demonstrated differences. The analysis of more potential confounders, as economic level (discussed in this section), distance from the centre, number of years of infertility would enrich the analysis if it is possible. A centre effect cannot be ruled out as potential / partial explanatory factor

8.    Conclusion

This study is really interesting, even if some points need to be addressed

9.    Recommendations

I recommend taking into account on my general questions.

Author Response

  1. Overall opinion

The paper concerns a relatively interesting question, with an almost perfect methodology, impressive results and is well written. However, there may be several explanations on the underlying mechanism that may be related to confounders which could not be addressed in the study. Moreover, several critical points need to be addressed to increase the demonstration. The references are very USA oriented (see below) and need to be enriched. Some methodological points can be discussed. However, after taking into consideration the reviewer’s remarks, the paper will surely deserves publication. – Thank you so much for the feedback.

  1. Introduction
  • Introduction is interesting, extended, with the potential mechanisms. However, many references are not very recent (11 of 27 before 2011), or rely on the same database (10 from SART), or from US data (n=24). Only 1 comes from outside and 2 from literature reviews. It is clear that the racial item is much more studied in USA than elsewhere, but the paper could be enriched with some other references as Wiltshire (2020), Chiang (2012), Mascarenhas (2019), Begum (2016), etc. could be included for discussion of racial differences. – Thank you so much for the feedback. We have incorporated as many of the aforementioned references that were able to find in order to incorporate data from outside the US. This can be seen in lines 86-87.
  • Define SAB before first use (I suppose spontaneous abortion) – This has been corrected
  1. Methods
  • Methods are extensively described and are an example of what has to be done in this matter. The inclusion/non-inclusion criteria are clear and a flow chart is given. – Thank you for this feedback.
  • Race affiliation needs to be explained for non-American readers and, if possible, discussed – We appreciate this point, race is explained in lines 118-120 along with a reference to CDC definitions.
  • Apparently, all initially included cycles were fresh and FET cycles (563730 cycles), and , then, only fresh cycles were included, merged with FET cycles issued from the finally included fresh cycles to make possible the cumulative rates calculations. This is ok but needs to be better explained. Moreover, it is necessary to explain why the merge was not done for 2016 fresh cycles, and how long after the index aspiration could the FET cycles could be merged (1, 2 years or more or only during the period 2014-2016). – We understand the confusion over the method of cycle linkage, this is all further explained in reference 14 which is another published study that we did on the same database. We only included a brief explanation to stay within the word limit for this journal.
  • Was the implantation rate only computed for the fresh cycle ? – Yes, implantation rates were only calculated for fresh cycles. We only included FET cycle data to cumulative live birth rate calculations.
  • Confounders: The most important were included, but a couple of ones that may have some relation with both racial category and outcome variable need to be at least discussed or included: economic status, distance from the ART centre. Finally, there is a potential bias if WNH people are of higher economic level and, thus, may access to centres with higher technicity. It may be difficult to analyse this factor, but that may play a role in the racial differences shown by authors. It is at least necessary to analyse a centre effect – Thank you for this valuable feedback. As mentioned in the discussion section, one limitation of the SARTCORS database is the lack of socioeconomic status data, we acknowledge that this is a shortcoming. As for distance to the ART centre, this data is also not included in the SARTCORS database and is also linked to socioeconomic status.

  1. Results

Results are well expressed. Tables 1 and 2 give a lot of statistically significant differences. However, there were apparently many covariables introduced in the multivariate analyses. The text is ambiguous between Material / methods and results sections – We appreciate this feedback; however, all of the covariables were needed to have a robust multivariable analysis. Each of the covariables, namely age, BMI, cause of infertility, history of past spontaneous abortions, use of ICSI and number of embryos transferred, are essential confounding factors which can influence live birth rates.

  1. Discussion
  • Discussion is well written, relatively extended. However, it is not easy to find a real explanation of the demonstrated differences. The analysis of more potential confounders, as economic level (discussed in this section), distance from the centre, number of years of infertility would enrich the analysis if it is possible. A centre effect cannot be ruled out as potential / partial explanatory factor – Thank you for this observation. As mentioned in the above comments, one limitation of the SARTCORS database is the lack of socioeconomic status data, we acknowledge that this is a shortcoming. As for distance to the ART centre, this data is also not included in the SARTCORS database, but is also linked to socioeconomic status. Furthermore, this study was meant to determine what outcomes disparities can be gleaned from the SARTCORS database. We fully acknowledge in the paper in lines 272-274 that socioeconomic factors may play an influence and future studies should assess this as a contributing factor in ART success.

  1. Conclusion

This study is really interesting, even if some points need to be addressed – We sincerely appreciate this feedback.

  1. Recommendations

I recommend taking into account on my general questions. – Thank you, we can incorporated your general concerns and questions into our paper.